# Using allocative efficiency analysis to inform health benefits package design for progressing towards Universal Health Coverage: Proof-of-concept studies in countries seeking decision support

Nicole Fraser-Hurt[1], Xiaohui Hou[1]*, Thomas Wilkinson[1], Denizhan Duran[1], Gerard J. Abou Jaoude[2], Jolene Skordis[2], Adanna Chukwuma[1], Christine Lao Pena[1], Opope O. Tshivuila Matala[1], Marelize Gorgens[1], David P. Wilson[3]

**1** The World Bank Group, Washington, DC, United States of America, **2** University College London Institute for Global Health, London, United Kingdom, **3** Bill & Melinda Gates Foundation, Seattle, Washington, United States of America

* xhou@worldbank.org

**Data Availability Statement:** The data underlying the Armenia analysis are available in a databook on:

## Abstract

### Background

Countries are increasingly defining health benefits packages (HBPs) as a way of progressing towards Universal Health Coverage (UHC). Resources for health are commonly constrained, so it is imperative to allocate funds as efficiently as possible. We conducted allocative efficiency analyses using the Health Interventions Prioritization tool (HIPtool) to estimate the cost and impact of potential HBPs in three countries. These analyses explore the usefulness of allocative efficiency analysis and HIPtool in particular, in contributing to priority setting discussions.

### Methods and findings

HIPtool is an open-access and open-source allocative efficiency modelling tool. It is pre-loaded with publicly available data, including data on the 218 cost-effective interventions comprising the Essential UHC package identified in the 3rd Edition of Disease Control Priorities, and global burden of disease data from the Institute for Health Metrics and Evaluation. For these analyses, the data were adapted to the health systems of Armenia, Côte d'Ivoire and Zimbabwe. Local data replaced global data where possible. Optimized resource allocations were then estimated using the optimization algorithm. In Armenia, optimized spending on UHC interventions could avert 26% more disability-adjusted life years (DALYs), but even highly cost-effective interventions are not funded without an increase in the current health budget. In Côte d'Ivoire, surgical interventions, maternal and child health and health promotion interventions are scaled up under optimized spending with an estimated 22% increase in DALYs averted–mostly at the primary care level. In Zimbabwe, the estimated gain was even higher at 49% of additional DALYs averted through optimized spending.

https://dataverse.harvard.edu/dataset.xhtml?
persistentId=doi:10.7910/DVN/LSX0BD Data for
Cote d'Ivoire and Zimbabwe largely came from
published government sources, as outlined in the
article. Access to the unpublished data is subject to
restrictions owing to privacy and ethics policies in
these countries. The data were made available to
the World Bank as technical partner in health
system analysis. Requests for access to the
specific unpublished data therefore need to be
made to the Ministry of Health and Child Care in
Zimbabwe and Ministry of Health and Public
Hygiene in Côte d'Ivoire. Inquiries can be made to
the article's corresponding author for further
guidance. The authors confirm that they did not
have special access privileges that others would
not have.

**Funding:** The authors received no specific funding
for this specific work.

**Competing interests:** The authors have declared
that no competing interests exist.

## Conclusions

HIPtool applications can assist discussions around spending prioritization, HBP design and primary health care transformation. The analyses provided actionable policy recommendations regarding spending allocations across specific delivery platforms, disease programs and interventions. Resource constraints exacerbated by the COVID-19 pandemic increase the need for formal planning of resource allocation to maximize health benefits.

## Introduction

Many countries now pursue some form of universal health coverage (UHC) policy objective, aligning with local and international commitments including the achievement of the Sustainable Development Goals [1, 2]. UHC implies that all people have access to the services they need, of sufficient quality to be effective, and without financial hardship. The services must be safe and effective, including access to affordable essential medicines and vaccines [3]. All people in need should be reached, including the most disadvantaged, and health service users should have improved financial protection, leading to a reduction in financial hardship experienced by households due to medical costs. Following the United Nations Sustainable Development Summit in 2015, many countries have adopted policy frameworks on universal access to essential health services and committed resources for the expansion of service delivery [4]. To reach UHC goals, countries need to both commit more money to health and get more health for the money [5, 6]. Country experiences demonstrate the importance of increased public financing for health care through compulsory, prepaid revenues for making UHC progress [7]. One key mechanism to address value for money is the development and provision of an evidence-based health benefits package (HBP) [8].

An HBP, sometimes called an essential or basic package of health services, is a defined set of services to be made available to everyone in the country [9]. Given that resources for health are limited, priority setting is inevitable when developing HBPs and decisions must be made on which interventions will be included. Priority setting for HBP design and progress toward UHC can involve balancing trade-offs between different UHC dimensions, affordability, as well as other locally relevant or valued criteria [7, 8, 10]. A common objective that guides HBP design is the efficient use of resources to maximize population health outcomes [11]. To meet this objective, a systematic and evidence-based consideration of the health gains achievable from many potential interventions is required.

The 3ʳᵈ edition of Disease Control Priorities (DCP3) proposed an evidence-based set of health interventions that are expected to provide good value for money in low- and middle-income settings, address a significant disease burden, and are feasible to implement in a large set of countries striving for UHC [12]. The 'model HBP' recommended by DCP3, called Essential UHC (EUHC), is comprised of 218 interventions delivered through different platforms considered representative of a typical health system: (a) population, (b) community, (c) health centers, (d) first-level hospitals and (e) referral hospitals. For the most resource constrained environments, DCP3 recommends the Highest Priority Package (HPP), which is a subset of 115 EUHC interventions selected based on a number of criteria [5]. The average costs and mortality averted of the EUHC and HPP interventions were estimated by the DCP3 initiative, which showed also that cost of implementation will vary by country context [5, 13, 14]. This is because the individual demographic and epidemiological situations, delivery costs, available budgets, and implementation structures vary from country to country. The DCP3 EUHC package is intended as a starting point to help guide the development of an HBP or review

existing provision, and it can be supplemented by data from the Tufts Registry [15] and the WHO UHC compendium [16].

While there is substantial global guidance for HBP design, countries must adjust recommendations based on local contexts and constraints. The Health Interventions Prioritization tool (HIPtool) has been developed to widen access to allocative efficiency analyses and assist countries in selecting, synthesizing and translating global evidence and HBP recommendations to their own health system's context. The tool, available as an online application (hiptool.org), uses a mathematical algorithm to determine allocations across defined health interventions which provide optimal health impact. HIPtool analyses are intended to initiate or contribute to the health resource allocation processes. To reduce the time and costs typically associated with allocative efficiency analyses, HIPtool draws on publicly available data sources including DCP3 health intervention data and the Institute for Health Metrics and Evaluation (IHME) global burden of disease database. Publicly available data can be customized in the tool to reflect country context, or replaced entirely with local data sources when available. The data are used to estimate the cost and health impact, in terms of disability-adjusted life years (DALY) averted, of various potential interventions and HBPs. The HIPtool optimization algorithm also estimates an optimized resource allocation within defined resource envelopes. The HIPtool is designed to assist decision makers in generating evidence to aid policy discussions around health intervention prioritization and HBP design.

The World Bank led early applications of HIPtool in three countries–Armenia, Côte d'Ivoire and Zimbabwe (Table 1). These case studies are the basis for the analyses presented in this paper.

This paper summarizes the experiences from HIPtool proof-of-concept studies in these three countries. With the increased demand for and availability of mathematical modeling for decision support in health, the paper aims to share the lessons learnt in the process.

## Methods

### Overall process

HIPtool applications constitute part of a broader set of processes and stakeholder consultations at country level, aiming to review public resource allocation, improve allocative efficiency, or support HBP definition. The steps of HIPtool implementation, including what data are required and expected model outputs, are shared with relevant stakeholders. Consensus is built around data sources and usage, stakeholder roles, secondary analyses, budget levels to be considered and optimization objectives. The tool facilitates a structured, evidence-driven dialogue focused on advancing UHC and defining a package of essential health interventions which is optimized to improve allocative efficiency in the following ways:

- HIPtool links data on intervention coverage, spending and cost-effectiveness to support discussions on what impact is being achieved with publicly financed interventions

- The optimization outputs suggest changes that could improve the impact of a set of interventions

- The ability of HIPtool to optimize different budget envelopes, can frame discussions on fiscal space for health and health financing

- The tool's options for weighting the relative importance of health impact, financial risk protection (FRP) and equity, contribute to discussions on how public service provision can provide value for money

Overall, stakeholder engagement throughout the HIPtool implementation process can create a space to discuss health reform options including tradeoffs between objectives of

**Table 1. Key data on the three country case studies.**

| Indicator | Armenia | Côte d'Ivoire | Zimbabwe |
|---|---|---|---|
| **Income classification** | Upper middle-income | Lower middle-income | Lower middle-income |
| **Region** | Europe & Central Asia | Sub-Saharan Africa | Sub-Saharan Africa |
| **Population size (2019)** | 2.97 million | 25.72 million | 14.65 million |
| **Domestic government health expenditure (% of GDP, 2018)** | 1.24% | 1.21% | 1.32% |
| **Health expenditure per capita (current US$, 2018)** | 422.28 | 71.88 | 140.32 |
| **Domestic general government health expenditure per capita (current US$, 2018)** | 52.11 | 20.71 | 39.25 |
| **Out of pocket expenditure as % of health expenditure (2018)** | 84.3% | 39.4% | 24.4% |
| **UHC effective coverage index***  | 62.4 | 43.0 | 54.5 |
| **Top 3 risk factors driving death and disability combined (2019)** | High blood pressure, tobacco, dietary risks | Malnutrition, air pollution, water-sanitation-hygiene | Malnutrition, unsafe sex, air pollution |
| **Outcomes:** | | | |
| **Life expectancy at birth,** | 74.9 years | 57.4 years | 61.2 years |
| **2018 (versus in 2000)** | (71.4 in 2000) | (49.6 in 2000) | (44.6 in 2000) |
| **Maternal mortality ratio, 2017** | 26/100,000 live births | 617/100,000 live births | 458/100,000 live births |
| **Under five mortality rate, 2019** | 11.8 /1,000 live births | 79.3 /1,000 live births | 54.6 /1,000 live births |
| **Health benefits package explicitly defined** | HBP since 1997. Covers primary health care services for all, and other services including hospital care and diagnostics for 30 socially vulnerable and special groups. The per-capita payment system aggregates services at the PHC level within the HBP | Universal health insurance recently launched with a benefits package under development | The publicly financed system has the National Health Services Package guiding resource allocation |
| **Efficiency and rationale for HIPtool application** | Efficiency has been gained through reduction in excess hospital capacity, and the HBP is being reviewed. The HIPtool application and other analytical support activities, led by the MOH with support from development partners, aim to contribute to a structured and systematic approach to HBP re-design | An actuarial study projected an annual financing gap of >$258 million in the new health insurance, doubling by 2028, if contribution and expenditure levels remained stable. This pointed to the need for an evidence-based prioritization mechanism with supportive analytics such as HIPtool use | Health financing analyses suggested that to achieve better health outcomes, spending efficiency should be improved. The HIPtool application was conducted to help identify areas or interventions that should be prioritized in order to improve spending efficiency |

Sources: https://data.worldbank.org/, http://www.healthdata.org, Duran et al. Cote d'Ivoire country report, unpublished; Internal government documents, unpublished.

* The Universal Health Coverage (UHC) effective coverage index aims to represent service coverage across population health needs and how much these services could contribute to improved health.

efficiency, equity and financial risk protection, reduce the potential for political capture of priority setting, and inform how limited health resources could be more efficiently allocated to progress towards UHC.

## HIPtool application

A HIPtool application is designed to help answer the following questions, subject to available data:

1. What are the current health interventions that are being implemented, what is the current level of spending on these interventions and what portion of disease burden does current spending avert?

2. How can spending be best allocated across essential interventions, health delivery platforms and disease programs to maximize DALYs averted, given the country's scale-up targets and intervention cost-effectiveness?

3. What health services or interventions outside of the optimized HBP would be cost-effective and important to deliver?

4. How do changes in available funding affect the interventions included in an optimized HBP and associated DALYs averted?

5. What are the gaps in data to enable effective priority setting across essential health interventions?

The default input parameters in HIPtool come from DCP3 [5], IHME's burden of disease study [17] and Tufts Cost Effectiveness Analysis Registry [15], among other sources. Implementation broadly follows five stages (Fig 1) and a detailed description of HIPtool is available in S1 Appendix. The first stage maps a country's existing HBP or publicly financed health interventions to the pre-loaded EUHC interventions (see S2 Appendix). Several HBP services delivered in a country might be grouped under one EUHC intervention, and a crosswalk might reveal mismatches between the two taxonomies that can be individually resolved. The

| | |
|---|---|
| 1. Definition of intervention list | - Map country's interventions to DCP3 list of essential UHC interventions<br>- Review of included interventions in terms of main health delivery platform and package/program the interventions belong to |
| 2. Data collection on population eligible or in need, current intervention coverage, target coverage and unit costs | - Collect and analyze data on population eligible or in need of an intervention and current coverage of each intervention among persons in need or eligible for the intervention<br>- Determine target coverage for each intervention based on policies<br>- Estimate the cost of providing the intervention per person and year |
| 3. Calculation and validation of current spending estimates | - Intervention spending is then automatically calculated based on population in need, current coverage level and unit cost<br>- Validate against National Health Accounts or other expenditure analyses |
| 4. Calculation of the current and maximum potential impact of each intevention | - Using HIPtool's methodology taking into account current and target coverage, cost-effectiveness and quality reduction factor<br>- Current disease burden averted by intervention spending is estimated by HIPtool |
| 5. Optimization and interpretation of model outputs | - Setting of optimization objective with weighting of cost-effectiveness, financial risk protection and equity scores<br>- Run different scenarios on allowable scale-up and budget levels<br>- Model iterations with stakeholder review |

**Fig 1. Overview of the five stages of the HIPtool application.** Source: Authors' summary. Note: DCP3 = Third edition of Disease Control Priorities, UHC = Universal health coverage.

selected interventions can either be grouped into 21 care packages as per DCP3 (S2 Appendix) or the country's own disease control programs. The pre-loaded information on care delivery platforms (community, health center, first-level hospital, referral hospital, population-based) can be edited based on the country's health system. Some interventions are defined across multiple levels of care, for example child delivery interventions are at community (management of low-risk labor symptoms and referral services), health center (basic emergency newborn and obstetric care, BEmNOC), and hospital (comprehensive emergency newborn and obstetric care, CEmNOC) levels of care.

The second stage of HIPtool applications involves using data on current and target coverage. This process involves triangulation, estimation and judgement calls, as intervention coverage among those in need or eligible is often not reported. Strategic documents and plans as well as key informants can inform scale-up coverage targets. Where unavailable, DCP3 coverage estimates and a target coverage of 80% are used [18]. Pre-loaded unit costs, defined as the annual cost to deliver an intervention per person, can then be automatically adjusted for each country using the DCP3 costing model online tool [18]. In general, local unit costs are used where available, and the pre-loaded, adjusted unit costs are used to fill the gaps.

The third stage uses data triangulation and mapping techniques to validate the bottom-up costing amounts with reported expenditure levels, such as health spending in National Health Accounts, adjusting cost parameters for inflation to the required year of analysis.

The fourth stage of a HIPtool analysis begins by considering incremental cost-effectiveness ratios (ICER) which are pre-loaded in the tool for most interventions based on DCP3 EUHC data. Where the default EUHC ICERs provided from the DCP3 data are not considered appropriate for the local health system context, users can recalibrate ICERs by, for instance, substituting ICERs with more recent estimates from the literature or the continuously updated Tufts Cost-Effective Analysis Registry [16]. Each intervention is also associated with FRP and equity scores based on the DCP3 methodology [19]. Once ICER estimates are selected, a 'quality reduction' factor is applied to reflect the fact that intervention effect sizes may be lower when implemented at scale in the health system context than that reported in the literature. A default 30% reduction in intervention effect is applied in HIPtool, which can be adjusted by users [20]. Intervention input data is then combined to estimate a 'maximum potential impact' (MPI) for each intervention, which is the estimated maximum impact an intervention could have on the causes of the disease burden it is linked to. The MPI parameter defines the ceiling of investment for each intervention during the optimization process, and is estimated assuming a linear relationship between spending, cost-effectiveness, and current and target nominal coverage (see S1 Appendix on how MPI is calculated).

The fifth stage requires the setting of the budget level to be optimized and the optimization objective. This is a user-defined combination of maximizing DALYs averted, equity scores and FRP. HIPtool's mathematical optimization function enables all the included health interventions to be assessed simultaneously, enabling the "whole of package" assessment that can be applied much faster than assessing individual interventions manually. The optimization module of the HIPtool is detailed in S1 Appendix. Model outputs on optimized spending patterns and health impact require extensive review and iterative analysis reflecting policy-relevant questions.

## Results

There were variations in how HIPtool was implemented in the three countries. There were also variations in the results and insights that could be gained at this proof-of-concept stage. We present them hereunder in brief (detailed technical country reports are in preparation or

published [21]). The calculations required for a country application can be viewed in a data book on the link http://hdl.handle.net/10986/35347 (Armenia example).

## Armenia: Rich data environment allows localization of HIPtool and provides policy-relevant outputs

**Policy context.** The HIPtool application was embedded in multiple other studies, including an actuarial costing of a benefits package, an assessment of strategic purchasing in the health sector and a projection of revenues from tax- and non-tax sources. All the studies focused on providing technical support for UHC and benefit package re-design in Armenia. The model was implemented in the first half of 2020 using 2019 HBP data and National Health Accounts (NHA) data up to and including 2018. The process was an inclusive one in which HIPtool Focal Points from various Ministry of Health (MOH) units, the State Health Agency, National Institute of Health, and World Bank Project Implementation Unit were instrumental in making data available in formats usable in the model. Modelling scenarios were discussed, including a higher budget scenario using a 38% increase in the current health budget level (associated with a hypothetical increase of the state budget to health from 5.8% to 8.0%). Focal Points were briefed about the tool and acted as key resource persons during the entire study to ensure appropriate data use and review. The State Health Agency collaborated closely by extracting large data sets from its ArMed e-health system. This system includes data on approximately 3,700 HBP service codes, which can be specific to defined social groups, geographical areas and refund entitlement levels. Multiple HBP service codes relate to per-capita services provided for free at primary health care (PHC) level pertaining to general medical practice, pediatrics, obstetrics/ gynecology and family medicine.

**Key takeaways and results from HIPtool implementation in Armenia.** *1. Intervention list*. The mapping and matching of Armenia's publicly financed health interventions to the pre-loaded EUHC interventions resulted in 135 Armenia UHC (AUHC) interventions which could enter the optimization analysis in the HIPtool. The step involved the review of approximately 3,700 coded HBP services and grouping them under EUHC interventions. The 135 retained interventions belonged to 19 of the DCP3's 21 care packages (excluding neglected tropical diseases and environmental health packages).

*2. Intervention coverage, target groups and unit costs*. This step replaced pre-loaded data with Armenia-specific data. For each AUHC intervention, 2019 coverage in the target population was estimated. The population 'in need' was defined using estimates on incidence or prevalence. The population 'eligible' drew on guidelines and demographic data (e.g. birth cohort, women of reproductive age). Epidemiological or demographic estimates were scaled down based on assumptions about the proportion of individuals requiring the intervention in a single year. For instance, diabetes screening in the target group is recommended to be done every three years, so the number of undiagnosed individuals in the eligible age range was divided by three to estimate the number requiring screening in 2019. Data were sourced from demographic and key population estimates, Armenia's vaccination schedule, disease burden data from IHME, national surveys [22–25] and other sources. Unit costs were calculated by summing all claims in the ArMed e-health system divided by the total number of claims in 2019, for all HBP services grouped under a specific AUHC intervention. Spends in four per-capita HBP codes were apportioned to nine PHC-level AUHC interventions using estimated target population, intervention coverage and unit cost. Similarly, spends for drug provision (two HBP codes for adults, two for children) were apportioned to a total of 15 AUHC interventions. MOH central-level spend for vaccines was mapped to vaccination-related AUHC interventions.

*3. Current spending.* The analysis included government and Global Fund health spending. The 135 AUHC interventions were associated with 56% of total government health expenditure as per the NHA (USD 93.809 million, of which 18.5% is within the capitation system). Another 44% of government health expenditure remained outside of AUHC interventions as the respective services could not be mapped to any of the HIPtool interventions, or the spend was linked to health worker salaries, infrastructure or consumables. Data triangulation across reported spending levels was imperfect as the analysis used 2019 HBP spending but 2018 NHA data, however, long-term NHA data was used to understand annual government health spending trends. Global Fund spending on AUHC interventions on HIV and TB was included in the optimization (leading to a total amount of USD 95.653 million in the optimization).

*4. Maximum potential impact.* This was calculated by projecting the intervention coverage levels in the HBP-eligible target groups to the Armenia population. Assumptions and judgement calls had to be made during this part of the process. For example, when an HBP service was provided for socially vulnerable and other special groups and emergency cases (e.g. surgical management of rectovaginal fistula), or socially vulnerable and other special groups, and children<18 (e.g. proctologic and reconstructive colon surgeries). Again, triangulation of disease burden, HBP claims, eligibility and survey/statistical data was essential to develop best possible estimates of population-level coverage. The tool's pre-loaded ICER values were generally used but reviewed and selectively replaced with more appropriate values from the literature, especially where ICERs were relatively low compared to the disease burden and reported spending. The pre-loaded FRP and equity scores were replaced by local scores, with highest scores given to the most expensive AUHC interventions and those targeting the most vulnerable groups (detailed in S3 Appendix).

*5. Optimization and interpretation of results.* Modeling scenarios varied several parameters to explore outcomes, including i) Budget: 2019 budget or a 38% higher budget; ii) Optimization constraints: Only 10% allowable coverage increase, or unconstrained; iii) Optimization weights for DALYs, FRP and equity: Equal, or high DALY weight, or low equity weight. The model outputs were overall similar for different weights given to DALYs, FRP and equity in the optimization step. Outputs were aggregated by each of the 21 EUHC care packages but not by service delivery platform due to many interventions using multiple platforms. It was estimated that the 135 interventions at 2019 budget and coverage averted 120,000 DALYs, and that optimized spending could increase the impact to 151,000 DALYs (Fig 2). If the budget available to fund the 135 essential AUHC interventions could be increased to the hypothetically feasible level (i.e. 38% increase on the 2019 health budget), an estimated 166,000 DALYs could be averted by the included interventions compared to the actual 2019 funding level and allocations.

## Côte d'Ivoire: Model implementation using pre-loaded DCP3 data provides directional results for investment in disease programs

**Policy context.**   HIPtool was implemented in Côte d'Ivoire between May 2018-October 2019, which overlapped with discussions in the country regarding ambitious health sector reforms. Reforms under discussion included the launch of a national health insurance scheme, the scale up of strategic purchasing and preparation of the Global Financing Facility (GFF) investment case to strengthen the health system and improve maternal and newborn health outcomes. Key national data came from the 2016 NHA and previous OneHealth Tool costing exercises (internal government documents, unpublished). The World Bank team worked closely with a Ministry of Health Core Team including representatives from the Cabinet (chief health financing advisor to the Minister of Health), Finance, Planning, Monitoring and

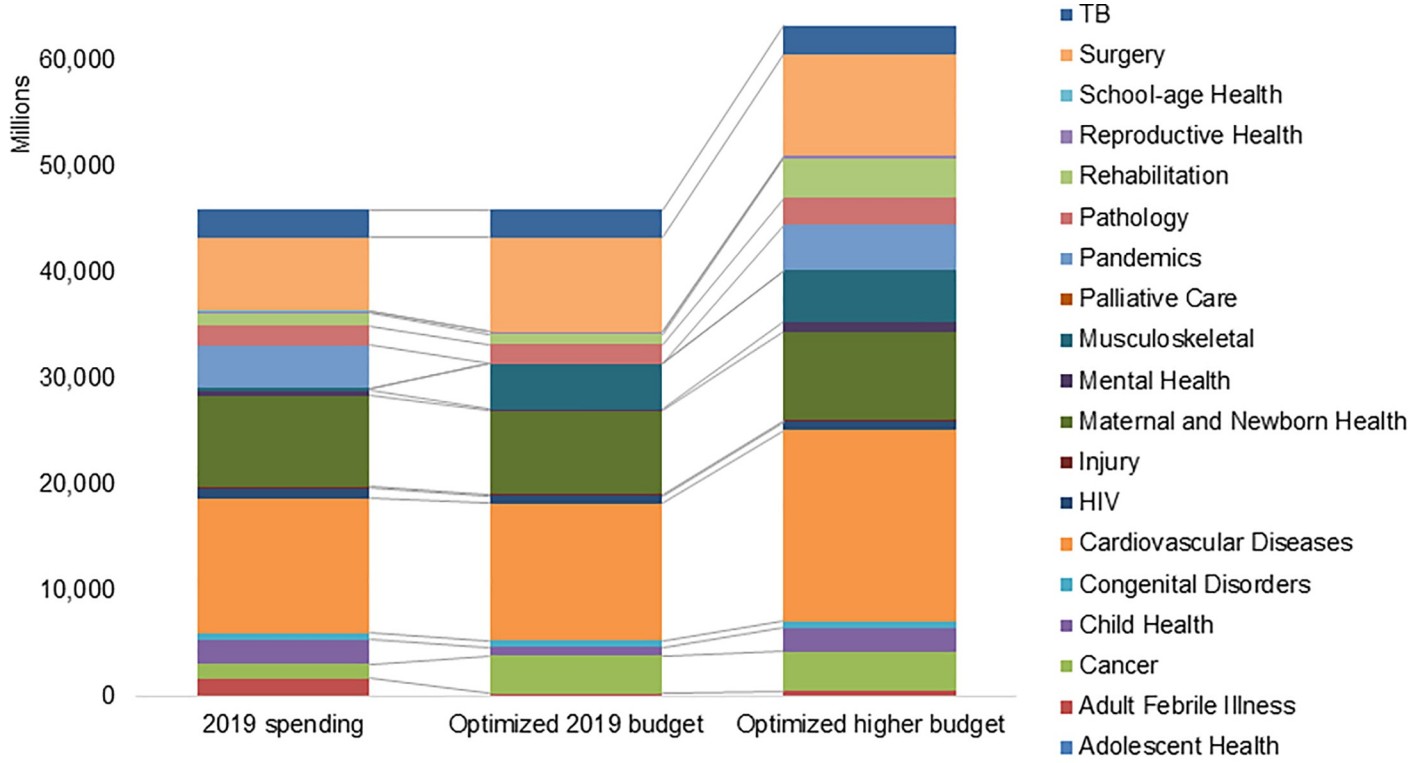

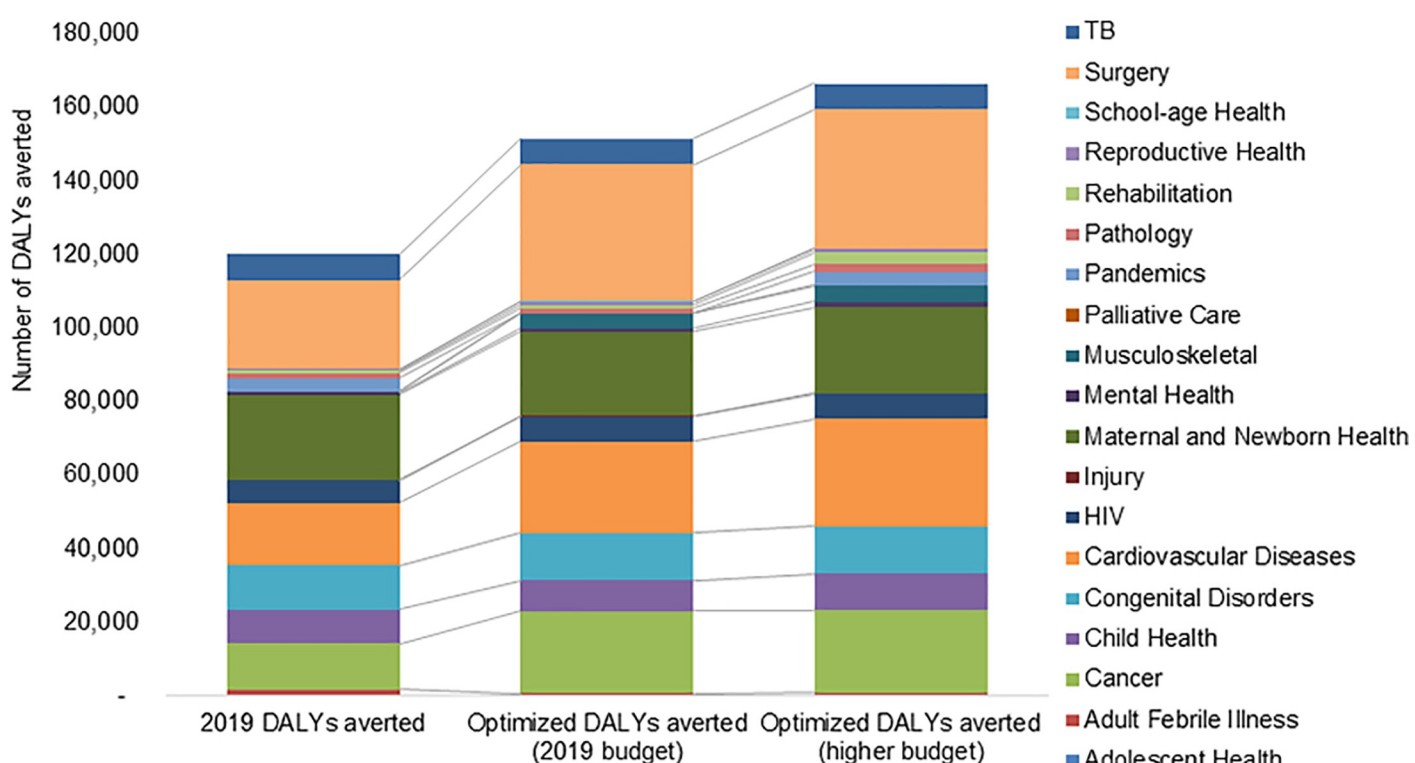

**Fig 2. Armenia model outputs on optimized resource allocation and health impact by EUHC care package (2019 vs. higher budget scenario).** A) Estimated spending. B) Estimated DALY impact. Source: Armenia HIPtool analysis. Note: Maximum allowable coverage increase = 90% (if 2019 baseline <90%, 95% (if baseline 90–94%, 100% (if baseline 95+%); child delivery interventions fixed at 2019 levels; Equal weights for DALYs, FRP and equity.

Evaluation, the newly launched health insurance agency, and health service directorates. The Directorate of Health Services overseeing the individual disease programs was also involved and provided essential intervention data. The Ministry of Health team also regularly liaised with the Directorate of Budget in terms of the results and how they could potentially be used to inform decision-making processes.

**Key takeaways and results from HIPtool implementation in Côte d'Ivoire.** *1. Intervention list.* Mapping existing services to the 218 EUHC interventions in HIPtool resulted in 179 interventions retained for the Côte d'Ivoire study. Given the absence of a single national HBP, stakeholder consultations and document review helped to arrive at the intervention list.

*2. Intervention coverage, target groups and unit costs.* For each intervention, coverage and target group data were collected considering levels of care. The tool's pre-loaded data were used for the interventions' default coverage levels and unit cost, unless the 2016 OneHealth assessment [26] the 2016 Multiple Indicator Cluster Survey [27] or other national data could be used. Only 26% of the interventions had data on local target populations and intervention coverage. Data on non-communicable diseases (NCDs) were especially scarce and international estimates were used. All pre-loaded unit cost data used were adjusted automatically in HIPtool. Key data sources were the population census [28], the Health Management Information System, the 2016 Multiple Indicator Cluster Survey [29] and IHME prevalence estimates.

*3. Current spending.* The analysis included all health spending irrespective of financing source, including out of pocket expenditures. Spending was calculated by multiplying the unit costs, target population and target coverage levels. A total of US$ 1.433 billion was included in the optimization and spending shares across disease programs was compared to 2016 NHA program spending data for validation. Significant mismatches were seen for NCD and surgical interventions, which constituted about a quarter of national health expenditures, but the NHA did not have detailed expenditure data at the intervention level which is needed for HIPtool optimization. In the final model, NCDs, malaria, maternal health and HIV were the disease programs with the largest expenditures adding up to almost 70% of all health spending, which was in line with the 2016 NHA spending patterns.

*4. Maximum potential impact.* This step linked the 179 interventions to the pre-loaded ICER values and reviewed the relationship between the burden of disease addressed by an intervention, as well as intervention spending and impact. The default 30% 'quality reduction' factor was applied, and the pre-loaded DCP3 data on intervention FRP and equity scores were maintained.

*5. Optimization and interpretation of results.* The optimization was run using target coverage levels. For interventions with coverage above 80%, target coverage was kept equal. For communicable disease interventions with coverage below 80%, target coverage was fixed at 80% following stakeholder discussions. For NCDs, target coverage was fixed at 30% following stakeholder discussions, due to very low baseline levels of provision. Optimization results were presented by care delivery platform (Fig 3) and national disease programs (S4 Appendix) rather than DCP3's care packages, to ensure relevance to existing planning and budget lines. The community platform gained most spending in the optimization (+68%), followed by the population-based platform (+31%) and the referral and specialist hospitals (+19%) (Fig 3A). Despite a 12% reduction in spending for each the health centre and the first level hospital platforms, the estimated DALY impacts were positive for all five platforms (Fig 3B). Shifts in spending for care delivery platforms ranged from a 68% increase for community-based

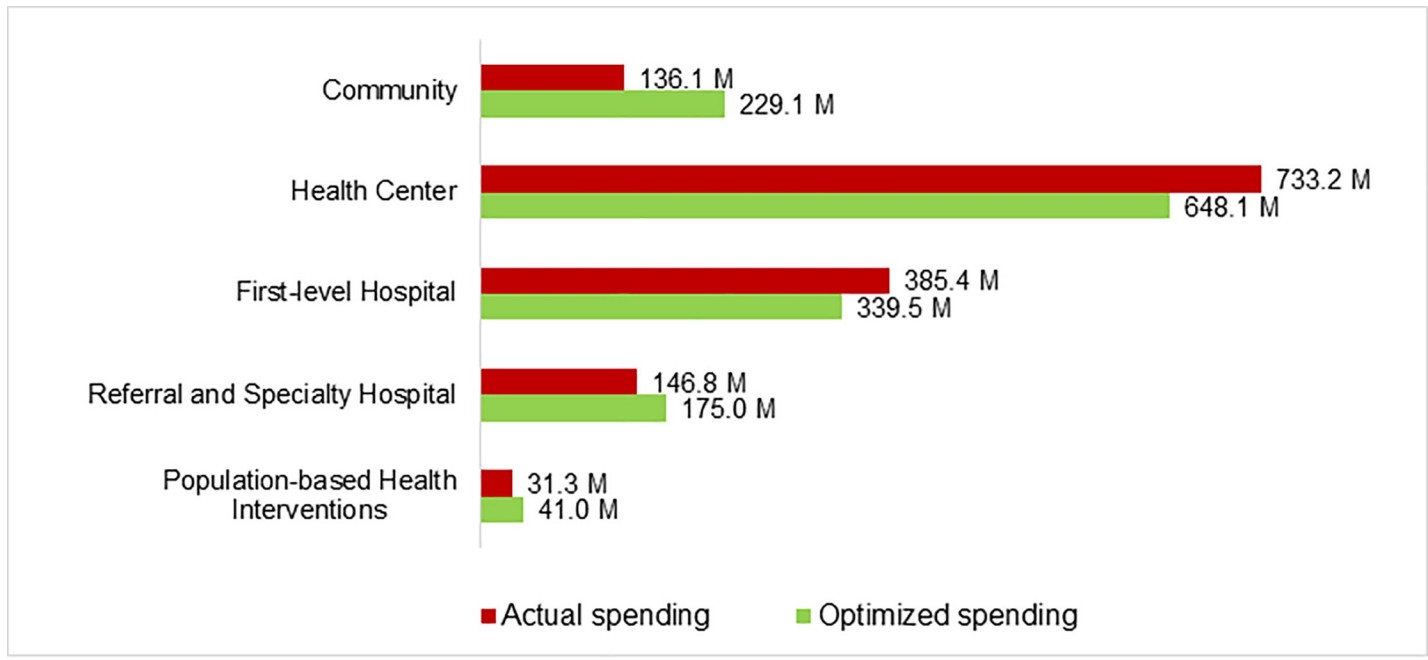

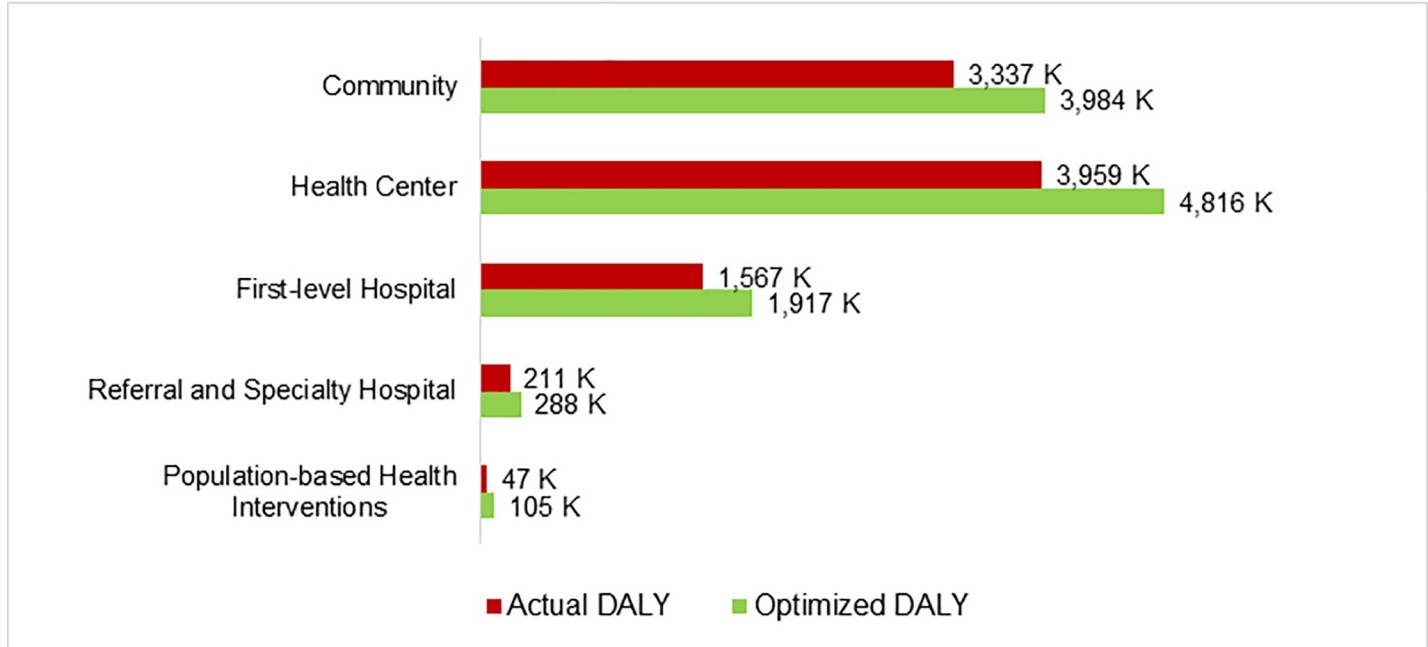

**Fig 3. Côte d'Ivoire model outputs on actual and optimized 2016 spending and impact by care delivery platform.** A) Estimated spending. B) Estimated DALY impact. Source: Côte d'Ivoire HIPtool analysis.

interventions to a 12% decrease for Health Centre and first level hospital. Under optimized spending scenarios, surgery, maternal health, child health and health promotion interventions were scaled up significantly. The DALYs averted increased for all disease programs due to a change in interventions included, from 9.1 million DALYs averted by current spending, to 11.1 million DALYs averted by the optimized allocation. Optimization significantly increased impact at the primary care level (i.e., community and health center), particularly for preventive

interventions in the community, going from 3.96 million DALYS to 4.82 million DALYs averted at the health center level, and going from 3.34 million DALYS to 3.98 million DALYs averted at the community level. Under the optimized scenario, although only 16% of the budget is disbursed at the community level, this spend averts 36% of DALYs. Overall, optimized spending at the community and primary care levels averted 79% of DALYs, for 45% of the budgeted spend. Analysis of interventions with the largest spending increases and DALY impact upon optimization (S4 Appendix) showed that four of the five interventions which, following optimization, see the most significant increase in DALYs averted are communicable disease-related, followed by care for fractures.

## Zimbabwe: Extensive data integration in HIPtool enables model implementation for decision support

**Policy context.**   In Zimbabwe, HIPtool was implemented during 2018–19 in response to several health financing studies that pointed to an urgent need to improve spending efficiency in Zimbabwe [30–33]. HIPtool implementation was guided and informed by interviews with key officials from the Ministry of Health and Child Care (MHCC), the Ministry of Finance and Economic Development, the Clinton Health Access Initiative and the World Health Organization, among other stakeholders. The analysis was based on 2016 fiscal year data, with expenditure, cost and budget data extracted from the 2016 Resource Mapping Report [34] the NHA Report [33] the National Health Strategy 2015–2020 [34] and MHCC's 2016 expenditure and appropriation account (unpublished). Also consulted were the 2015 National Health Service (NHS) report (unpublished), WHO Global Health Observatory [35], Demographic and Health Surveys [36], Multiple Indicator Cluster Surveys [37], UNAIDS HIV data [38], published, peer-reviewed and grey literature, agencies' databases including UN Populations Division and UNICEF, and the Global Burden of Disease database for disease prevalence [19].

**Key takeaways and results from HIPtool implementation in Zimbabwe.**   *1. Intervention list.* Most of the NHS package was successfully mapped to 176 of the 218 EUHC interventions pre-loaded in HIPtool, using NHS protocols and consulting with local experts.

*2. Intervention coverage, target groups and unit costs.* Like the other country applications, included interventions were linked to populations in need or eligible, coverage levels and unit cost. Data was drawn from a broad range of sources and pre-populated HIPtool data was used whenever required. Local unit costs from the National Health Strategy 2015–2020 mostly related to MCH interventions. The resource mapping study provided data on HIV and malaria spending. Otherwise, the pre-populated EUHC intervention unit costs were used, following adjustment, and inflation to 2016.

*3. Current spending.* Intervention spending was estimated by combining unit cost and annual utilization estimates for each of the interventions. Available expenditure and budget data were then used to compare and validate aggregated intervention expenditure estimated with HIPtool [39]. Of the estimated US$980 million spent on health from domestic government and external sources, US$672 million (69%) were included in the optimization. The remaining US$308 million were not included in the optimization, primarily because no direct link could be established between spending and burden of disease. This applied for instance to health systems strengthening, palliative care, prevention and relief of refractory suffering, resuscitation with basic measures and NCD behavior change communication. Therefore, for the 19 interventions or expenditure activities that were not included in the optimization, associated spending was fixed and remained unchanged in the analysis.

*4. Maximum potential impact.* The interventions included in the optimization were linked to the pre-loaded ICER values, but the team also consulted additional DCP3 data to find the

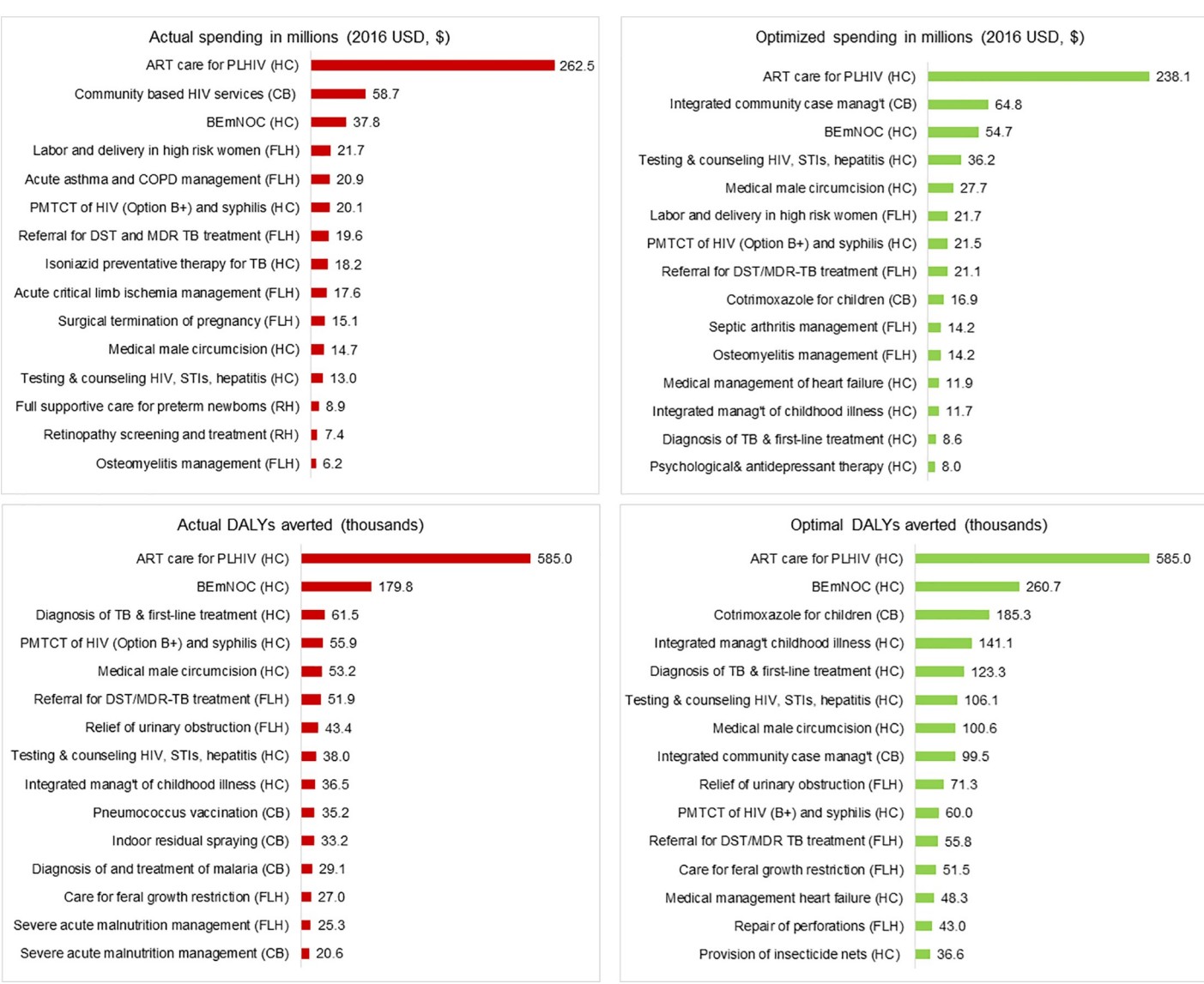

**Fig 4. Zimbabwe model outputs on actual and optimized 2016 spending and impact by intervention.** CB = Community-based, HC = Health Center, FLH = First Level Hospital, RH = Referral and Specialty Hospital. A) Highest expenditure interventions (2016 actual versus optimized). B) Most impactful interventions (2016 actual versus optimized). Source: Zimbabwe HIPtool analysis.

most appropriate values where ranges of ICERs from different contexts were available. All ICERs were adjusted by the default 30% 'quality reduction' factor.

*5. Optimization and interpretation of results.* Full weighting was assigned to health maximization, prioritizing ICER values over the FRP and equity scores of the included interventions. Results from the HIPtool optimization analysis suggested that several interventions should be allocated more funding, with the largest increases in the 2016 budget level allocated to integrated community case management, HIV and STI testing, cotrimoxazole for children and medical male circumcision (Fig 4A). Maternal and child health interventions remained critical in the optimized allocation and most priority interventions were delivered at the community and primary health center level. TB interventions remained a strong area of focus, involving diagnosis and first-line treatment at the health center level and MDR-TB diagnosis and

treatment at a first-level hospital. None of the interventions for which additional spending was prioritized were at the referral and specialty hospital level or population-wide. The model estimated that 938,000 DALYs could be averted through optimized allocations in addition to the estimated 1.9 million DALYs averted by 2016 spending (Fig 4B). Under optimized spending, health center interventions would account for 55% of all DALYs averted, and a further 583,000 (20%) and 366,000 (13%) DALYs could be averted through community and first-level hospital interventions, respectively (S4 Appendix). The impact of referral and specialty hospital interventions decreases by 9,000 DALYs under optimized allocations of spending. The optimized allocation of 2016 national health spending is therefore estimated to increase the impact of all but one platform of care, and generate cost-savings through first-level hospital interventions that yield greater impact with a 25% spending reduction.

## Discussion

There are substantial data on current and future trends in countries' disease burdens, health service costs and coverage, and the expected impact of specific health interventions. Decision makers in health have an unprecedented body of evidence to draw on but navigating such complex data can be challenging. HIPtool was developed to address this challenge by combining available data on intervention cost, coverage and effectiveness across all major disease programs of a health system, within a mathematical optimization algorithm. Mathematical optimization tools use defined algorithms to process multiple data points to identify highest (or lowest) values across multiple objectives within a defined constraint. Applied to HBP design, mathematical optimization approaches can identify an "optimal" series of health interventions within a health budget constraint that will independently or jointly maximize mutually exclusive objectives such as health, equity, and financial risk protection.

While the outputs of the HIPtool cannot answer all the questions about how to attain UHC or how to formulate the specifics of an HBP, the results presented here demonstrate how a model like HIPtool can provide directional results on high-impact, priority interventions and programs in different settings and therefore indicate allocatively efficient investments.

One strength of applying a model like HIPtool is that it can provide the structure for a *systematic, evidence-based and transparent process* that includes stakeholder engagement, review and appraisal of the evidence and extensive discussion of policy options. All three proof-of-concept studies demonstrated that the application of HIPtool informed in-country policy discussions on defining HBPs and identifying key spending priorities to bridge gaps to UHC and support primary health care transformation. As such, HIPtool can be a key input to deliberative and data-driven priority setting processes as countries define or update their benefits packages.

In Armenia, HIPtool could be localized to a large extent thanks to the richness of recent local data. Tool implementation drove secondary analysis of HBP claims data and generated 'unit prices' for each AUHC intervention. The first ever health system-wide estimation of coverage levels highlighted gaps of concern. With a few exceptions like diabetes treatment, it highlighted insufficient public financing of services addressing chronic diseases and conditions. A more efficient HBP would provide better cover for conditions linked to old age, given the rapidly aging population with high rates of Alzheimer's disease, falls, hearing loss, cataract, cancers, low back pain, and type 2 diabetes [40]. Given the high burden of ischemic heart disease and stroke, it was no surprise that optimization pointed to cardiovascular disease prevention and management as a consistent top priority for funding. The significant increased investment in musculoskeletal and cancer packages in the optimized allocation suggests there are currently missed opportunities for health impact. The analysis identified specific areas to

increase value for money in the HBP, such as better cover for physiotherapy, long term management of heart and vascular diseases, primary prevention of osteoporosis, palliative care, HPV vaccination for schoolgirls, diagnosis and treatment of colorectal cancer, and several rehabilitation interventions. However, it also demonstrated the limited impact of any HBP with highly constrained levels of public funding.

In Côte d'Ivoire, HIPtool implementation served as an entry point for priority setting discussions around the benefits package and available health interventions and has been welcomed by policymakers at both health and finance ministries. As there was no defined benefits package in Côte d'Ivoire at the time of analysis and dissemination, the implementation of HIPtool also informed discussions on the ground pertaining to evidence-based mechanisms to define and update the benefits package at different levels of care. Actionable policy implications included the need to increase spending on maternal health, child health and surgical interventions, to reallocate resources within disease programs towards more cost-effective interventions, and to increase spending on community interventions, particularly for NCDs, due to the significant potential to avert DALYs. The insights from the process have contributed to discussions on budgeting, the new health sector strategic plan and GFF investment case processes, as well as building demand for routine data, especially for NCDs. Findings from the HIPtool analysis also have implications for the health insurance and strategic purchasing arrangements, as well as donor coordination and reallocation of donor budgets. The study marks the beginning of possible longer-term technical and political engagement to improve the allocative efficiency of health spending in Côte d'Ivoire.

In Zimbabwe, the use of HIPtool, combined with other technical analyses, helped shape the policy dialogue on how allocative efficiency can be increased in the health sector while bearing in mind local economic, political and implementation realities. The model provided further evidence of the high impact that well-funded HIV and TB treatment programs have, but also showed the large share of health spending for HIV/AIDS, and the need to assess the allocative efficiency of individual HIV interventions given the low resource levels for other priority programs. Consistent with global literature [41], the most impactful and prioritized interventions were those delivered at lower platforms of care, such as the Primary Health Centers (PHCs). Given that spending on MCH and NCD-related interventions increased under an optimized allocation, an emphasis on integrated care emerged as an important step to improving spending efficiency, such as integrated community case management and BEmNOC at PHCs. A cost-effective HBP would shift public spending from hospitals to community and PHC platforms of care, with community health workers and PHC staff spearheading integrated care models. Similarly, NCD interventions could be more broadly delivered at the PHC and community levels to improve cost-effectiveness. Zimbabwe has identified strengthening budget formulation processes as a key and urgent reform. The HIPtool implementation process represented an important step toward better use of local and international evidence for resource allocation across disease programs. This is especially important in the context of the current fiscal situation.

HIPtool can be flexible in defining the total budget to be allocated, which can be tailored to the country context. In both Zimbabwe and Côte d'Ivoire, development partner financing is a significant source of overall health financing, thus the interventions prioritization was analyzed including both public and development partner financing in health. The tool is also efficient in calculating and comparing various budget scenario analyses, normally within a few minutes once the underlying data are complete and validated, to assist health policy makers in exploring the impact of budget change and trade-offs across averted DALYs, equity and FRP.

Through all three applications of HIPtool, useful lessons were learnt about the importance of stakeholder collaboration during tool implementation. Engaging stakeholders early and

often, identifying a champion, seeking consensus, and joint data identification and vetting were all essential to a well-supported process. Ad-hoc consultation with data owners and the securing of expert support for customized analyses helped ensure the quality of model inputs. Stakeholders' interest in capacity building on allocative efficiency modeling and in future tool use signaled an understanding of the important role analytics can play in attaining value-for money HBPs and UHC goals. This understanding and institutional knowledge gained through these HIPtool applications has the potential to improve the quality and speed of future analyses.

HIPtool offers an interactive way of engaging with stakeholders. Once the tool data is adjusted and deemed appropriate, users can test policy scenarios and update model inputs continuously. In Côte d'Ivoire and Zimbabwe, a significant value of this proof-of-concept application was that discussions on benefits packages started taking place for the first time. National strategic plans in both countries tend to be designed and executed in a disease-specific manner, without sector-wide prioritization. These discussions, if continued, have the potential to improve allocative efficiency and equity across the system, particularly with the launch of evidence-based health insurance products. In Armenia, the lack of regulation guiding the systematic revision of the HBP including clarity on what factors should be considered, stakeholders to be involved, and mechanisms for transparency, have so far hampered HBP re-design in line with UHC goals [42]. In all three countries, HIPtool applications nurtured a process infrastructure for future discussions on the prioritization of health resources, with a focus on costs and coverage across the entirety of the health system.

## Limitations

The studies had several limitations: First, since HIPtool adopts a health system perspective, it is unable to capture cross-sectoral benefits that lie outside of the health budget. For example, effects such as gains in productivity or school attendance cannot be captured. This means that the positive impact of an HBP, as estimated by HIPtool, is likely to be an underestimate of the full cross-sectoral impact. Second, since there is a trade-off between usability and flexibility, there are many highly complex interactions between diseases and health interventions that cannot be captured in full, as these would require more data than are available in most country contexts. As a consequence, overlaps and synergies between the interventions included in HIPtool are not considered, and must be accounted for in the data being uploaded or parameterisation of intervention MPIs. Also, only the static first-order impacts are currently incorporated into the analyses. Third, HIPtool is not a disease modelling tool and therefore does not currently account for disease progression and infectiousness, which would introduce substantial complexity and data needs. Instead, it builds on the best existing projections of disease burden and studies of intervention effects in terms of mortality and DALYs averted. Fourth, the tool does not provide a timeline for intervention scale-up, which is instead explored by running multiple scenarios. As such the analyses did not include a dynamic temporal dimension or adjustments in effectiveness or cost at scale, but rather emphasize the possibility of reallocating resources at any given point. Fifth, as with any mathematical model, there are numerous data limitations. For instance, the demographic data might not reflect true population sizes in a country due to migration which is for instance significant among men in Armenia. The analyses performed by HIPtool are only as valid as the data entered into it. However, HIPtool revealed important data gaps and "made a case" for increased focus and investment in the collection of data supporting decision-making. Moving forward, the UHC compendium [43] recently released by WHO provides a comprehensive database of interventions and associated data that can be used as inputs in HIPtool and address a number of existing data challenges.

In all three model applications, the largest uncertainties were related to the ICERs. The latter are based on a large global effort but may not reflect the cost-effectiveness of the locally delivered intervention, and the availability of ICERs as point estimates rather than ranges does not enable uncertainty analyses. Again, HIPtool analyses can point to key ICER values which should be refined through local studies, however, in environments where localized evidence and analytics are limited, it may be challenging to develop valid local parameters on a scale to inform all interventions in the HBP. This is a particularly important component in the current optimization algorithm where the ICER value provides the basis for the estimation of effect size and efficiency. Future iterations of the HIPtool algorithms can disassociate the effect and efficiency parameters, but this is dependent on disaggregated cost and effects data being made available for EUHC interventions [44]. However, the release of the UHC compendium is also likely to enable improved future iterations of the HIPtool optimisation algorithm. Finally, HIPtool is not a costing or budgeting tool, and is intended only to contribute one component in the overall process of determining an effective HBP. Political, logistical, and other considerations need to be considered outside HIPtool to determine what HBPs are feasible. Any HBP will need to be carefully costed and the implications for implementation fully considered. These considerations are outside the scope of HIPtool, but are critical for the successful adoption of an HBP. The above limitations continue to inform methodological improvements in the HIPtool. Particular areas for methodological improvement in the tool include integrating a more consistent intervention taxonomy, automation of calculations for current population in need and current coverage (while maintaining functionality to manually adjust estimates) and mechanisms for importing a greater range of estimates for intervention costs effectiveness estimates. The HIPtool limitations also highlight improvements that countries can take to generate data that usefully inform priority setting. Such data systems need to collect data not only on inputs (such as staffing and commodities) and outputs (such as number of outpatient appointments), but integrate with epidemiological and population data to generate an understanding of how interventions are addressing existing demand. In addition, localized economic evaluation and costing of a greater number of interventions will considerably improve the applicability of results and the priority setting process in general. Nevertheless, while the methodology is being further improved, the current HIPtool outputs can inform high-level discussions on allocative efficiency and provide an entry point into more specific and comprehensive analyses in collaboration with other existing tools.

## Conclusions

There is a clear need to set health sector priorities, especially now that COVID-19 has further constrained fiscal space and increased the burden on health systems globally. A recent global health investment analysis suggested that where governments maintain pre-COVID 19 trends in health spending, the share of government resources flowing to health will have to increase on average by more than 11% above pre-COVID levels [45]. A growing number of countries are defining HBPs to allocate limited resources, illustrated here by the case of Cote d'Ivoire, or review their HBPs (Armenia and Zimbabwe cases). A HIPtool application within a country, serves to demonstrate the gains in terms of health impact and related health system expenditure, that can be achieved by taking an allocative efficiency conceptual approach to HBP design. While allocative efficiency tools such as HIPtool cannot replicate complex system realities due to data and tool limitations, they allow testing of policy options and can provide useful directional results. Equally important, they can give structure to the health policy process, mobilize stakeholders and centralize an array of health data, if well implemented. The HIPtool findings may also have implications for donor coordination and effective allocation of donor

funding. Decision support tools like HIPtool, combined with other technical analyses and consideration for political and implementation realities, will provide the necessary evidence to improve health resource allocations, transition from passive to strategic purchasing, and achieve more health for health spending.

## Supporting information

**S1 Appendix. HIPtool technical specifications.**
(PDF)

**S2 Appendix. HIPtool interventions by delivery platform and DCP3 care package, with respective global burden of disease causes addressed.**
(PDF)

**S3 Appendix. Local equity and financial risk protection scores (Armenia).**
(PDF)

**S4 Appendix. Additional optimization results.**
(PDF)

## Acknowledgments

We thank the stakeholders in Armenia, Cote d'Ivoire and Zimbabwe for their participation in the analyses at all stages from design to completion, and the peer reviewers of the three country studies for providing comments that helped improve the quality of this manuscript. We also thank Lara Goscé, Shepherd Shamu, Chenjerai N. Sisimayi, Laurence Lannes, Cliff Kerr, Hassan Haghparast-Bidgoli, Marianna Koshkakaryan, Lusine Yengibaryan for their invaluable contributions to the respective country studies. The findings, interpretations, and conclusions expressed in this paper are entirely those of the authors. They do not necessarily represent the views of the World Bank and its affiliated organizations, or those of the Executive Directors of the World Bank or the governments they represent.

## Author Contributions

**Conceptualization:** Xiaohui Hou, Jolene Skordis, Marelize Gorgens, David P. Wilson.

**Data curation:** Denizhan Duran.

**Formal analysis:** Nicole Fraser-Hurt, Xiaohui Hou, Thomas Wilkinson, Denizhan Duran, Gerard J. Abou Jaoude, Adanna Chukwuma.

**Methodology:** Gerard J. Abou Jaoude, David P. Wilson.

**Project administration:** Xiaohui Hou, Adanna Chukwuma, Christine Lao Pena, Opope O. Tshivuila Matala.

**Software:** Xiaohui Hou, Gerard J. Abou Jaoude, Jolene Skordis.

**Supervision:** Xiaohui Hou, Jolene Skordis.

**Validation:** Thomas Wilkinson, Denizhan Duran.

**Visualization:** Nicole Fraser-Hurt, Thomas Wilkinson.

**Writing – original draft:** Nicole Fraser-Hurt, Xiaohui Hou.

**Writing – review & editing:** Nicole Fraser-Hurt, Xiaohui Hou, Thomas Wilkinson, Denizhan Duran, Gerard J. Abou Jaoude, Jolene Skordis, Adanna Chukwuma, Christine Lao Pena, Opope O. Tshivuila Matala, Marelize Gorgens, David P. Wilson.

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
