## [Decision Letter · Decision Letter 0]

4 Sep 2021

PONE-D-21-15273

Using allocative efficiency analysis to inform health benefits package design for progressing towards Universal Health Coverage: Proof-of-concept studies in countries seeking decision support

PLOS ONE

Dear Dr. Hou,

Thank you for submitting your manuscript to PLOS ONE. After careful consideration, we feel that it has merit but does not fully meet PLOS ONE’s publication criteria as it currently stands. Therefore, we invite you to submit a revised version of the manuscript that addresses the points raised during the review process.

We look forward to receiving your revised manuscript.

Kind regards,

M. Mahmud Khan

Academic Editor

PLOS ONE

Journal Requirements:

3. We note that you have referenced (Duran et al. Cote d’Ivoire, Internal government documents, unpublished, Ministry of Health and Child Care Zimbabwe, Expenditure and appropriation account 2016 (unpublished, Results from the Health Interventions Prioritization Tool (unpublished]) which has currently not yet been accepted for publication. Please remove this from your References and amend this to state in the body of your manuscript: (Duran et al. Cote d’Ivoire, Internal government documents, unpublished, Ministry of Health and Child Care Zimbabwe, Expenditure and appropriation account 2016 (unpublished), Results from the Health Interventions Prioritization Tool (unpublished])as detailed online in our guide for authors http://journals.plos.org/plosone/s/submission-guidelines#loc-reference-style.

Reviewers' comments:

Reviewer's Responses to Questions

**Comments to the Author**

1. Is the manuscript technically sound, and do the data support the conclusions?

Reviewer #1: Yes

Reviewer #2: Partly

2. Has the statistical analysis been performed appropriately and rigorously? 

Reviewer #1: Yes

Reviewer #2: Yes

3. Have the authors made all data underlying the findings in their manuscript fully available?

Reviewer #1: No

Reviewer #2: Yes

4. Is the manuscript presented in an intelligible fashion and written in standard English?

Reviewer #1: Yes

Reviewer #2: Yes

5. Review Comments to the Author

Reviewer #1: This article addresses critical resource allocation issues within developing countries health sector. The authors did a great job in explaining the framework, data, and analytic part. They also did a good job addressing limitations of the study. There are couple of items that, if addressed, can strengthen the article.

1. Authors use governmental statistics to construct their analysis. In developing countries use of this data can be problematic (as authors indicate) due to reliability. For example, population estimates usually are overestimates since a large number immigrates seasonally or permanently without it being reflected in the updated numbers due to several reasons. Armenia is a classical case in this regards as a large number of (male) population usually seasonally works in Russia.

2. it will be helpful to use similar graphics style in displaying results. Also, will be helpful to the reader to have comparative graphs (if possible ) to compare outcomes for three countries.

Reviewer #2: The topic of the paper is very important. I find the main problem with presentation of the paper, which may be difficult for the readers to follow. The following comments might be useful for improvement.

Comment: Authors wrote (page 12, just before table 2) that the HIPtool optimization algorithm also estimates an optimized resource allocation within defined resource envelopes.

Can you detail this somewhere in the paper?

Comment: Involving stakeholders is a good process for selecting the variables/data etc. It would be more useful to know who those stakeholders were (researchers, development partners, or any others).

Comment: You should give a heading of the first column in table 1.

Comment: While it is well-describe what HIPtool does, I miss a clear description of how it does all the steps. For instance, the authors wrote the questions that HIPtool is able to answer. But it is unclear how the tool does so.

Comment: Figure 1 is a good presentation. A simple example should be developed in the paper (might be fictitious), relating figure 1 so that the readers can follow the complex calculations (country cases).

Comment: The authors should make the concept of optimization clearer in the text and detail the optimization process that they used. They should describe more clearly for the readers how they address allocative efficiency in this very context (in methods and findings as well as in results).

Comment: The authors wrote about the limitations. It would be great if they explain how this tool and analysis are useful despite such limitations. They should also guide from their experience (of this paper/work) what data and methodological improvement are needed so that HIPtool or any similar ones can be used more appropriately.

Comment: Some of the text is appendices may be summarized (while keeping the appendices as they are) and put in the main manuscript so that they readers can read the article independently.

Comment: Conclusions are too general. The authors can additionally connect to the findings of three country cases.

6. PLOS authors have the option to publish the peer review history of their article (what does this mean?). If published, this will include your full peer review and any attached files.

Reviewer #1: No

Reviewer #2: No

---

## [Author Response · Author response to Decision Letter 0]

16 Oct 2021

Reviewers’ comments and responses

1. Is the manuscript technically sound, and do the data support the conclusions?

Reviewer #2: Partly

 We hope that our revisions to the manuscript respond to the concerns of this reviewers regarding the technical quality of our report.

3. Have the authors made all data underlying the findings in their manuscript fully available?

Reviewer #1: No

The default data is available in http://hiptool.org (the weblink to the application is now provided in the introduction), which contains the DCP3 interventions, estimation of unit costs and ICER values. We have also provided a weblink allowing readers to access the publicly available, read-only Excel workbook from Armenia which illustrates the data inputs and calculations/estimations, as well as scenarios in an application. Data for Zimbabwe and Cote D’Ivoire can be available through formal request. 

Review Comments to the Author and Responses

Reviewer #1: This article addresses critical resource allocation issues within developing countries health sector. The authors did a great job in explaining the framework, data, and analytic part. They also did a good job addressing limitations of the study. 

 Thank you very much.

There are couple of items that, if addressed, can strengthen the article.

1. Authors use governmental statistics to construct their analysis. In developing countries use of this data can be problematic (as authors indicate) due to reliability. For example, population estimates usually are overestimates since a large number immigrates seasonally or permanently without it being reflected in the updated numbers due to several reasons. Armenia is a classical case in this regard as a large number of (male) population usually seasonally works in Russia.

 We thank the reviewer for this comment and agree. We have made an addition in the limitations section: “For instance, the demographic data might not reflect true population sizes in a country due to international migration which is significant among men in Armenia.”

2. it will be helpful to use similar graphics style in displaying results. Also, will be helpful to the reader to have comparative graphs (if possible) to compare outcomes for three countries.

We have harmonised the colours in Figures 3 and 4 (red/green) and adjusted a label in Figure 3. Regarding the choice of graphs in the main article, we would like to keep the current selection as the intention is to present the types of outputs the HIPtool can provide. This reflects the nature of the report, which is about the local applications rather than comparison of model outputs between the three countries. 

Reviewer #2: The topic of the paper is very important. 

 Thank you very much.

I find the main problem with presentation of the paper, which may be difficult for the readers to follow. The following comments might be useful for improvement.

Comment: Authors wrote (page 12, just before table 2) that the HIPtool optimization algorithm also estimates an optimized resource allocation within defined resource envelopes. Can you detail this somewhere in the paper?

 We agree with the reviewer that the reader should be referred to more explanation about the optimization steps, and have made an insert in the methodology section to the section ‘optimization module’ in Appendix 1 of the supplementary material.

Comment: Involving stakeholders is a good process for selecting the variables/data etc. It would be more useful to know who those stakeholders were (researchers, development partners, or any others).

We thank the reviewer for this important point, as this type of analysis entirely relies on effective stakeholder involvement. Given they are somewhat country-specific, we have mentioned them in the ‘policy context’ sections of the three countries, rather than upfront when describing the general approach. For Armenia, the key stakeholders we had listed were the HIPtool Focal Points from various Ministry of Health units, the State Health Agency, National Institute of Health, and World Bank Project Implementation Unit. For Cote d’Ivoire, the key stakeholders listed are a Ministry of Health Core Team including representatives from the Cabinet (chief health financing advisor to the Minister of Health), Finance, Planning, Monitoring and Evaluation, the newly launched health insurance agency, the Directorate of Health Services Directorate of Budget. In Zimbabwe, the main stakeholders have equally been mentioned in this section as the Ministry of Health and Child Care (MHCC), the Ministry of Finance and Economic Development, the Clinton Health Access Initiative and the World Health Organization.

Comment: You should give a heading of the first column in table 1.

 Thank you, we have completed the table with an insert labelling this column.

Comment: While it is well-describe what HIPtool does, I miss a clear description of how it does all the steps. For instance, the authors wrote the questions that HIPtool is able to answer. But it is unclear how the tool does so.

We thank the reviewer for this comment on how the model works. All main aspects are described in Appendix 1, including how the impact model operates to link interventions to disease burden and effect, how the concept of effective and maximal effective coverage is applied, the optimization module itself, as well as the equity and financial risk protection modules. We have included the main equations and trust that this information can help the reader follow how the tool works. The focus of this article is on tool implementation in different country contexts.

Comment: Figure 1 is a good presentation. A simple example should be developed in the paper (might be fictitious), relating figure 1 so that the readers can follow the complex calculations (country cases).

We thank the reviewer for the positive comment on figure 1. In order to illustrate the calculations carried out in a country application, we have inserted a link which enables the reader to access the Armenia HIPtool data book (Results section, first paragraph). 

Comment: The authors should make the concept of optimization clearer in the text and detail the optimization process that they used. They should describe more clearly for the readers how they address allocative efficiency in this very context (in methods and findings as well as in results).

 We appreciate this suggestion and have made several explanatory additions in the text. 

Comment: The authors wrote about the limitations. It would be great if they explain how this tool and analysis are useful despite such limitations. They should also guide from their experience (of this paper/work) what data and methodological improvement are needed so that HIPtool or any similar ones can be used more appropriately.

 We agree with the reviewer that readers may be interested to hear how the tool’s limitations inform its methodological improvements. We have therefore added several sentences under the limitations heading. The addition highlights key improvements such as the use of a more consistent intervention taxonomy, the automation of certain calculations while maintaining functionality to manually adjust estimates, and better data import functionalities. We also reflect on what countries can do to strengthen the process and generate data that usefully inform priority setting. 

Comment: Some of the text is appendices may be summarized (while keeping the appendices as they are) and put in the main manuscript so that they readers can read the article independently.

We thank the reviewer for this suggestion. With the additions made especially on the optimization step (see above), we believe that some of the technical detail about the tool is now better summarized in the main text. We are open to further additions but want to be sensitive to the journal guidelines on concise presentation and discussion of findings. The main article has now a length of approximately 7,000 words. 

Comment: Conclusions are too general. The authors can additionally connect to the findings of three country cases.

We thank the reviewer for the suggestion. We have only made small changes in the conclusions, which we would like to keep at a high level. The discussion already covers the countries’ individual and shared insights from the assessment.

---

## [Editor Report · Decision Letter 1]

8 Nov 2021

Using allocative efficiency analysis to inform health benefits package design for progressing towards Universal Health Coverage: Proof-of-concept studies in countries seeking decision support

PONE-D-21-15273R1

Dear Dr. Hou,

We’re pleased to inform you that your manuscript has been judged scientifically suitable for publication and will be formally accepted for publication once it meets all outstanding technical requirements.

Kind regards,

M. Mahmud Khan

Academic Editor

PLOS ONE
---

## [Editor Report · Acceptance letter]

16 Nov 2021

PONE-D-21-15273R1 

Using allocative efficiency analysis to inform health benefits package design for progressing towards Universal Health Coverage: Proof-of-concept studies in countries seeking decision support 

Dear Dr. Hou:

I'm pleased to inform you that your manuscript has been deemed suitable for publication in PLOS ONE. Congratulations! Your manuscript is now with our production department. 

Kind regards, 

on behalf of

Dr. M. Mahmud Khan 

Academic Editor

PLOS ONE